# Improving Gaussian Splatting with Localized Points Management

## Abstract

Point management is critical for optimizing 3D Gaussian Splatting models, as point initiation (e.g., via structure from motion) is often distributionally inappropriate. Typically, Adaptive Density Control (ADC) algorithm is adopted, leveraging view-averaged gradient magnitude thresholding for point densification, opacity thresholding for pruning, and regular all-points opacity reset. We reveal that this strategy is limited in tackling intricate/special image regions (*e.g.*, transparent) due to inability of identifying all 3D zones requiring point densification, and lacking an appropriate mechanism to handle ill-conditioned points with negative impacts (*e.g.*, occlusion due to false high opacity). To address these limitations, we propose a **Localized Point Management** (LPM) strategy, capable of identifying those error-contributing zones in greatest need for both point addition and geometry calibration. Zone identification is achieved by leveraging the underlying multiview geometry constraints, subject to image rendering errors. We apply point densification in the identified zones and then reset the opacity of the points in front of these regions, creating a new opportunity to correct poorly conditioned points. Serving as a versatile plugin, LPM can be seamlessly integrated into existing static 3D and dynamic 4D Gaussian Splatting models. Experimental evaluations validate the efficacy of our LPM in boosting a variety of existing 3D/4D models both quantitatively and qualitatively. Notably, LPM improves both static 3DGS and dynamic SpaceTimeGS to achieve state-of-the-art rendering quality while retaining real-time speeds, excelling on challenging datasets such as Tanks & Temples and the Neural 3D Video dataset.

## 1 Introduction

Neural rendering has emerged as a generalizable, flexible, and powerful approach for photorealistic novel view synthesis (NVS) of any camera poses (Mildenhall et al., 2021), underpinning a wide variety of applications in augmented/virtual/mixed reality (Deng et al., 2022b), robotics (Yang et al., 2023), and generation (Poole et al., 2022), among more others. For example, taking a learning-based parametric idea, Neural Radiance Fields (NeRFs) (Mildenhall et al., 2021) implicitly represent the scene radiance of any complexity using neural networks (*e.g.*, MLPs), without the tedious requirements of model handcrafting for accounting the scene variations in geometry, texture, illumination. However, their view rendering is inefficient computationally due to heavy ray sampling, thus suffer in scaling to high-resolution content applications and large scale scene modeling (Tancik et al., 2022; Turki et al., 2022).

Recently, 3D Gaussian Splatting (3DGS) (Kerbl et al., 2023) has come as an alternative with explicit representation, much faster model optimization and real-time neural rendering. The process begins by initializing a set of 3D Gaussian points using Structure from Motion (SfM) (Snavely et al., 2006). This is followed by optimizing the parameters of these points through view reconstruction loss, resulting in a view output generated with differentiable splatting-based rasterization. However, the point initialization is often distributionally non-optimal, leading to issues such as under-population (e.g., insufficient points) or over-population (e.g., excessive points) in the 3D space. Consequently, a point management mechanism, such as Adaptive Density Control (ADC), is necessary during optimization. However, we identify several limitations with ADC: (i) Thresholding the average gradient to determine regions for point densification often overlooks under-optimized points. For instance, larger Gaussian points typically have lower average gradients and may frequently appear

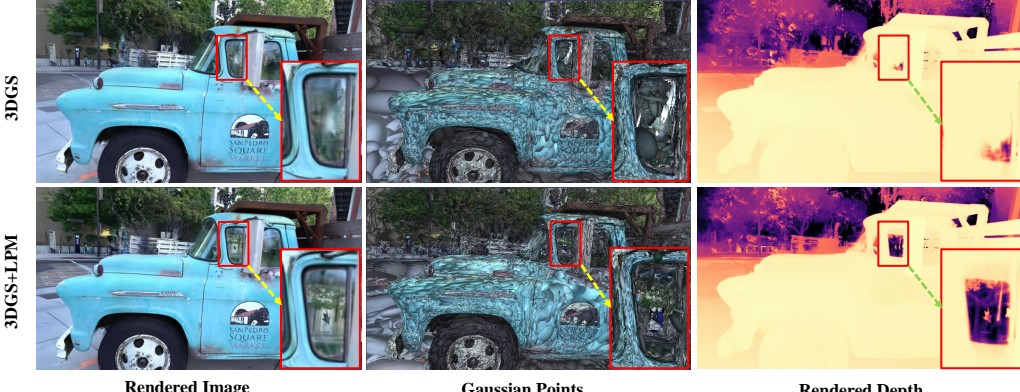

**Figure 1:** Visualization of points behavior. 3DGS produces ill-conditioned Gaussians (red box) that occlude other valid points, resulting in noticeably incorrect depth estimation. **LPM** handles these ill-conditioned points to reduce negative impacts and further calibrate the geometry.

across various views in screen space. (ii) Point sparsity complicates the addition of sufficient and reliable points needed to comprehensively cover the scene. (iii) Mis-optimized Gaussian points can have detrimental effects, such as occluding other valuable points and leading to incorrect depth estimates (see erroneous placements on windows in Fig. 1).

To overcome the aforementioned limitations, in this paper we propose a novel *Localized Point Management* (LPM) approach. Our idea is intuitive – identifying those 3D Gaussian points leading to rendering errors. Thus we start with an image rendering error map of a specific view. To obtain the error contributing 3D points, we leverage the region correspondence between different views via feature mapping, subject to the multiview geometry constraint. For each pair of corresponded regions, we cast the rays through them at their respective camera views in the cone shape, and consider their *intersection* as the error source zone. Within each such zone, we consider two situations: (1) At presence of points, we further apply point densification at a lower threshold to complement the original counterpart locally; (2) In case no point due to point sparsity, we add new Gaussian points. Concurrently, we reset the opacity of points with high opacity estimates that are located in front of these zones, as they can significantly affect view rendering. This provides an opportunity to correct potentially ill-conditioned points while tuning the newly added ones in the subsequent optimization. To minimize model expansion, we prune the points by opacity in a density-aware manner.

We summarize the ***contributions*** below: (1) Through in-depth analysis, we have identified several limitations in the standard point management mechanism used in Gaussian Splatting that impede model optimization. (2) We present ***Localized Point Management*** (LPM) for these issues by identifying error-contributing 3D zones and implementing appropriate operations for point densification and opacity reset. (3) Extensive experiments validate the benefits of our LPM in improving a diversity of existing 3D and 4D Gaussian Splatting models in novel view synthesis on both static and dynamic scenes.

## 2 RELATED WORK

**Neural Scene Representations** has always been an important direction in novel view synthesis. Previous methods allocate neural features to structures such as volume (Lombardi et al., 2019; Sitzmann et al., 2019), texture (Thies et al., 2019), and point cloud (Aliev et al., 2020). The pioneering work of NeRF (Mildenhall et al., 2021) proposes integrating neural networks with 3D volumetric representations to convert a 3D scene into a learnable density field, enabling high-quality novel view synthesis without requiring explicit modeling of the 3D scene and illumination. Later on, numerous works emerge to boost the quality and efficiency of volume rendering, (Barron et al., 2021; Xu et al., 2022; Barron et al., 2023) refine the point sampling strategy in ray marching, some some advanced works (Barron et al., 2022; Wang et al., 2023) reparameterize the scene to produce a more compact

representation. Additionally, regularization terms (Deng et al., 2022a; Yu et al., 2022) can be incorporated to constrain the scene representation, resulting in a closer approximation to real-world geometry. Despite their high-quality representational performance, these methods are typically computationally inefficient for view rendering due to the extensive ray sampling required and the use of Multi-Layer Perceptrons (MLPs) to represent the scene, complicating the computation and optimization of any point within the scene. To address this, several works have proposed novel scene representations aimed at accelerating the rendering process. These representations replace MLPs with sparse voxels (Liu et al., 2020), hash tables (Müller et al., 2022), or triplanes (Chen et al., 2022), significantly enhancing rendering speed. However, real-time rendering remains challenging due to the inherent complexity of the ray marching strategy in volume rendering.

**Gaussian Splatting** represents a recent advancement in novel view synthesis, enabling real-time high-quality rendering. It contributes to splatting-based rasterization by computing pixel colors through depth sorting and $\alpha$-blending of projected 2D Gaussians, thereby avoiding the complex sampling strategies of ray marching and achieving real-time performance. It is precisely due to its real-time high-quality rendering capabilities that 3DGS has been applied to various domains, including autonomous driving, content generation (Tang et al., 2023), and 4D dynamic scenes (Li et al., 2023; Wu et al., 2023; Yang et al., 2024), among others. Despite these advancements, 3DGS still has some drawbacks, such as the storage of Gaussians and handling multi-resolution, and so on. Several works have enhanced 3DGS by improving Gaussian representation, including techniques such as low-pass filtering (Yu et al., 2023), multiscale Gaussian representations (Yan et al., 2023), and interpolating Gaussian attributes from structured grid features (Lu et al., 2023). However, these works often overlook the importance of point management, specifically Adaptive Density Control, which is typically applied during optimization to address issues like under-population or over-population in the 3D space. Only a few works have focused on point management. For example, GaussianPro (Cheng et al., 2024) directly tackles densification limitations, bridging gaps from SfM-based initialization. Pixel-GS (Zhang et al., 2024) proposes a gradient scaling strategy to suppress artifacts near the camera. Additionally, (Rota Bulò et al., 2024) introduces an auxiliary per-pixel error function to implicitly supervise point contributions.

Although these methods improve densification, they are still unable to identify all 3D zones that require point densification and lack a proper mechanism to handle ill-conditioned points with negative impacts. Here, we propose a novel approach, Localized Point Management, capable of identifying error-contributing zones with greatest demand for both point addition and geometry calibration.

## 3 METHOD

### 3.1 PRELIMINARIES: 3D GAUSSIAN SPLATTING

Gaussian Splatting builds upon concepts from EWA (Zwicker et al., 2001) splatting and proposes modeling a 3D scene as a collection of 3D Gaussian points $\{G_i \mid i = 1, \ldots, K\}$, rendered through volume splatting. Each 3D Gaussian $G$ is defined by the equation:

$$G(x) = e^{-\frac{1}{2}(x-\mu)^T \Sigma^{-1} (x-\mu)},$$

where $\mu \in \mathbb{R}^{3 \times 1}$ represents the mean vector, and $\Sigma \in \mathbb{R}^{3 \times 3}$ denotes its covariance matrix. To maintain the positive semi-definite nature of $\Sigma$ during optimization, it is represented as $\Sigma = RSS^T R^T$, with the orthogonal rotation matrix $R \in \mathbb{R}^{3 \times 3}$ and the diagonal scale matrix $S \in \mathbb{R}^{3 \times 3}$.

To render an image from a specific viewpoint, the color of each pixel $p$ is determined by blending $N$ ordered Gaussians $\{G_i \mid i = 1, \ldots, N\}$ that overlap $p$, using the formula:

$$c(p) = \sum_{i=1}^{N} c_i \alpha_i \prod_{j=1}^{i-1} (1 - \alpha_j),$$

where $\alpha_i$ is derived by evaluating a projected 2D Gaussian from $G_i$ at pixel $p$ combined with a learned opacity for $G_i$, and $c_i$ is the learnable, view-dependent color modeled using spherical harmonics in 3DGS. Gaussians that influence $p$ are arranged in ascending order based on their depth from the current viewpoint. Employing differentiable rendering techniques allows for the end-to-end optimization of all Gaussian attributes through training view reconstruction.

**Point management** Since existing 3DGS variants start by initializing 3D Gaussian points using Structure from Motion (SfM), the points are often coarse and non-optimal in space. During optimization, a point management mechanism, Adaptive Density Control (ADC), is typically applied to manage point distribution issues. Specifically, thresholding the average gradient is used to decide on point densification. For each Gaussian point $G_i$, 3DGS tracks the magnitude of the positional gradient $\frac{\partial L_\pi}{\partial \mu_i}$ across all rendered views, which is then averaged to a quantity $T_i$. During each training iteration, if the gradient $T_i$ surpasses a predefined threshold, it considers this point as inadequately representing the corresponding 3D region. With the scale of the Gaussian as the size measure, a large Gaussian will be split into two, while a small one leads to point cloning.

However, this commonly used ADC strategy is unable to identify all the 3D zones with the underlying need for point densification. This is becuase, often the local complexity of scene geometry varies significantly, which beyond the reach of any single-value based thresholding. Besides, there is lacking of a proper mechanism to handle ill-conditioned points with negative impacts (e.g., wrong opacity values estimated during training with points distributed here and there).

## 3.2 Localized Gaussian Point Management

To address the aforementioned issues, we introduce a novel model agnostic point management approach, *Localized Point Management* (LPM), which leverages multiview geometry constraints to identify error contributing 3D points, with the guidance of image rendering errors. This approach can be seamlessly integrated with existing 3DGS models without the need for architectural modification. As illustrated in Figure 2, we begin with an image rendering error map for a specific view. Under the multiview geometry constraint, the corresponding regions in the referred view are matched via feature mapping. For each pair of corresponding regions, we then cast rays through them from their respective camera views in a cone and identify their intersection as the error source zone. Within each zone, we perform localized point manipulation.

**Error map generation** To accurately localize those zones in the 3D space that require point densification and geometry calibration, we initiate our process by rendering the current view image through the splatting of 3D Gaussians. This is followed by generating an error map (Figure 2(a)) for this specific view against the grounth-truth image using an error function (Li et al., 2023).

**Error contributing 3D zone identification** To project this rendering error back to the 3D space, we leverage the region correspondence between different views under multiview geometry constraints. This involves the following two key steps.

*(i) Cross-view region mapping* We select a neighboring view as the referred image. Following LightGlue (Lindenberger et al., 2023) that predicts a partial assignment between two sets of local features extracted from two view images $A$ and $B$. Each feature consists of sets of 2D features position $\{F_i \mid (x_i, y_i) \in [0, 1]^2\}$, normalized by the image size. The images $A$ and $B$ contain $M$ and $N$ local features. LightGlue outputs a set of correspondences $\mathcal{M} = \{(i, j)\} \subseteq A \times B$. Since the 2D rendering error regions in the current view may not all appear in the referenced image, we select the paired region $(R_e, R_e')$ (Figure 2(b)) through the matching points. Additionally, this paired region undergoes multiview adaptive adjustments based on the error map throughout the optimization process.

*(ii) 2D-to-3D projection* After obtaining the paired regions with render errors, we project each 2D error region to the 3D space via multiview geometry constraints. Specifically, we cast the rays $\mathcal{C}$ in cone shape for region $R_e$ from the camera's center of projection $o$ along the direction $d$, which aligns with the pixel's center (Figure 2(c)). The apex of this cone is located at $o$, and its radius at the image plane. Hence, $o + d$ is parameterized as $\mathcal{C}$. The radius $r_{Cone}$ is set to match the radius of the smallest circumscribed circle of the 2D plane error region, creating a cone on the 3D space that can trace the Gaussian points contributing to the 2D error region. Concurrently, a corresponding cone, denoted as $\mathcal{C}'$, belong to region $R_e'$ is similarly projected. Subsequently, we compute the intersection points of these rays. In order to regionalize these points, we directly use a smallest sphere that can contain these points as error source 3D zone $R_{zone}$.

**Points manipulation** Recall that in existing 3DGS, points management only relies on the view-averaged gradient magnitude $\tau$ to determine point densification *globally*. In addition to this, we

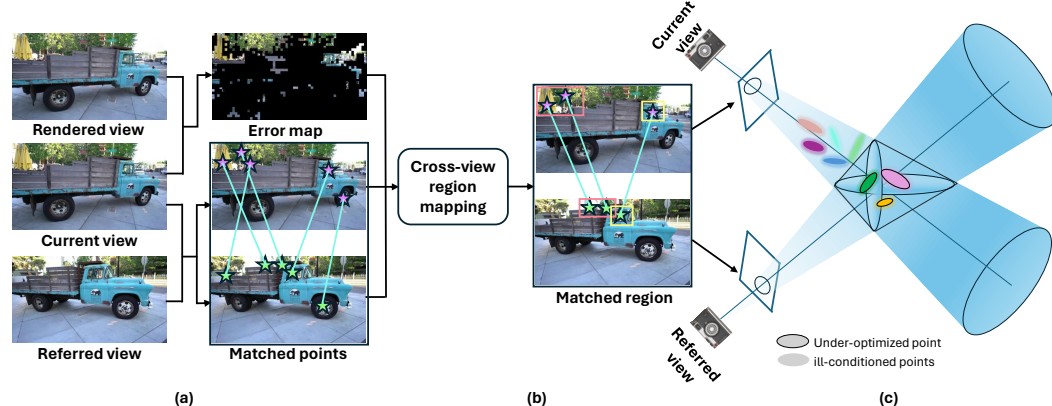

Figure 2: Overview of our Localized Point Management (LPM). (a) We start with an image rendering error map *versus* the current view (the ground-truth). Concurrently, matching points are identified between the current view and a refereed view sampled as an adjacent view via off-the-shelf feature mapping. (b) Subsequently, cross-view region mapping is then employed to locate the correspondence region in the refereed view. (c) For each pair of corresponded regions, we cast the rays through them at their respective camera views in the cone shape, and consider their *intersection* as the error source zone. The final step involves identifying under-optimized or ill-conditioned points within these zones, where under-optimized/empty places are densified, and ill-conditioned points are reset.

*further* perform localized points addition and geometry calibration within the identified error source 3D zone $R_{zone}$. For the point addition, we consider two common situations: (1) In the presence of points, we apply point densification to locally complement the original counterparts. We set a lower threshold to select the points that need densification, aiming to enhance the geometric details. The densification rule is consistent with 3DGS, but it focuses on local 3D zones that need it most. Specifically, for small Gaussians, our strategy involves cloning the Gaussians while maintaining their size and repositioning them along the positional gradient to better capture emerging geometrical features. Conversely, larger Gaussians situated in areas of high variance are split into smaller points to more accurately represent the underlying geometry. (2) In cases of point sparsity, we add new Gaussian points at the center of the 3D zone.

In the context of $\alpha$-blending in 3DGS, if the points at the forefront of the identified 3D zone $R_{zone}$ have the highest opacity, they may occlude valid points, leading to incorrect depth estimation, as shown in Figure 1. To deal with such issues, we treat these points as potentially ill-conditioned. We reset these points to provide an opportunity for correction, further calibrating the geometry.

To minimize model expansion, we adaptively prune points based on their opacity values, starting from low to high. The number of points pruned is determined by the density of points in the zone. This strategic reduction ensures that our point management remains cost efficient and adaptive to the evolving needs of the scene representation.

## 4 EXPERIMENTS

**Datasets and metrics** We conducted an extensive evaluation using both static and dynamic scenes derived from publicly datasets. For static scenes, our approach was applied to a total of 11 scenes as specified in the 3DGS framework (Kerbl et al., 2023), which includes nine scenes from Mip-NeRF360 (Barron et al., 2021), two from Tanks&Temples (Knapitsch et al., 2017), and two from DeepBlending (Hedman et al., 2018). In the context of dynamic scenes, our approach was tested across six scenes from the Neural 3D Video Dataset (Li et al., 2022b).

To evaluate novel view synthesis performance, we followed standard protocols by selecting one out of every eight images as test images, with the remaining used for training in static scenes. For each

dynamic scene within the Neural 3D Video Dataset, one view was designated for testing while the others were allocated for training purposes. Evaluation metrics included the peak signal-to-noise ratio (PSNR), structural similarity index measure (SSIM), and the learned perceptual image patch similarity (LPIPS), which are broadly recognized standards in the field.

**Baselines and implementation**   Vanilla 3D Gaussian Splatting (3DGS) (Kerbl et al., 2023), 2D Gaussian Splatting (2DGS) (Huang et al., 2024), Mip Gaussian Splatting (MipGS) (Yu et al., 2023), PiexlGS (Zhang et al., 2024) and SpacetimeGS (STGS) (Li et al., 2023) were selected as our main baselines for their established art performance in novel view synthesis. For the static 3D benchmark, we also recorded the results of Mip-NeRF360 (Barron et al., 2021), iNGP (Müller et al., 2022) and Plenoxels (Fridovich-Keil et al., 2022) as in (Kerbl et al., 2023). For the Dynamic 4D benchmark, we performed system comparison, such as DyNeRF (Li et al., 2022a), K-planes (Fridovich-Keil et al., 2023) and so on. In alignment with the approach described in 3DGS an STGS, our models were trained for 30k iterations across all scenes, following the same training schedule and hyperparameters. In addition to the original Gaussian densification strategies used in 3DGS and SpaceTime Gaussian, we also performed localized points management, including addition, reset, and pruning. We maintained the same thresholds for splitting and cloning points as in the original 3DGS and SpaceTime Gaussian. For point matching, we performed offline extraction to save computational cost. All experiments were conducted on an RTX 3090 GPU with 24GB of memory.

## 4.1 MAIN RESULTS

Table 1: Comparison of various methods across different scenes on the Mip-NeRF 360 dataset, Tanks&Temples and Deep Blending. * indicates the retrained model from the official implementation. Bold represents best, underline indicates second best.

| Method | Mip-NeRF 360 | | | Tanks&Temples | | | Deep Blending | | |
|---|---|---|---|---|---|---|---|---|---|
| | PSNR | SSIM | LPIPS | PSNR | SSIM | LPIPS | PSNR | SSIM | LPIPS |
| Plenoxels | 23.08 | 0.625 | 0.463 | 21.08 | 0.719 | 0.379 | 23.06 | 0.795 | 0.510 |
| INGP-Big | 25.59 | 0.699 | 0.331 | 21.92 | 0.745 | 0.305 | 24.96 | 0.817 | 0.390 |
| Mip-NeRF 360 | 27.69 | 0.792 | 0.237 | 22.22 | 0.759 | 0.257 | 29.40 | 0.901 | 0.245 |
| 3DGS | 27.21 | 0.815 | 0.214 | 23.14 | 0.841 | 0.183 | 29.41 | 0.903 | 0.243 |
| 3DGS* | 27.47 | 0.816 | 0.216 | 23.67 | 0.849 | 0.177 | 29.55 | 0.904 | 0.245 |
| 3DGS* + **LPM** | 27.59 | 0.820 | 0.216 | 23.83 | 0.850 | 0.181 | **29.76** | 0.908 | 0.241 |
| 2DGS* | 27.15 | 0.808 | 0.246 | 23.58 | 0.832 | 0.185 | 29.35 | 0.899 | 0.262 |
| 2DGS* + **LPM** | 27.42 | 0.817 | 0.228 | 23.65 | 0.848 | 0.180 | 29.52 | 0.903 | 0.240 |
| MipGS* | 27.51 | 0.817 | 0.210 | 23.69 | 0.852 | 0.173 | 29.58 | 0.910 | 0.242 |
| MipGS* + **LPM** | 27.70 | 0.821 | 0.210 | 23.82 | 0.851 | 0.180 | 29.61 | 0.910 | 0.241 |
| PixelGS* | 27.54 | 0.819 | 0.203 | 23.75 | 0.850 | 0.175 | 29.58 | **0.920** | 0.220 |
| PixelGS* + **LPM** | **27.80** | **0.830** | **0.190** | **24.02** | **0.856** | **0.173** | 29.65 | 0.910 | **0.196** |

**Results on static 3D datasets**   The quantitative results (PSNR, SSIM, and LPIPS) on the Mip-NeRF 360 and Tanks & Temples datasets are presented in Tables 12. We retrained the 3DGS model (referred to as 3DGS*) as it yields better performance compared to the vanilla 3DGS and its variants. Our approach achieves results comparable to the state-of-the-art on the Mip-NeRF360 dataset and further enhances all 3DGS based method using our point management technique. Additionally, LPM improve vanilla 3DGS and PiexlGS to set new state-of-the-art results on the Mip-NeRF 360, Tanks & Temples datasets and Dep Blending, effectively capturing more challenging environments (*e.g.*, light effects, transparency). These results quantitatively validate the effectiveness of our method in improving the quality of reconstruction.

In Figures 3, we present a comparison between 3DGS (Kerbl et al., 2023) and 3DGS + LPM, focusing on both appearance and depth. A variety of improvements can be observed, particularly in challenging cases such as light effects, completeness at a distance. Our LPM significantly reduces artifacts in specific regions on top of 3DGS, particularly in the tree at the second. These regions require more points for accurate population, leading to a more precise and detailed reconstruction. Additionally, the tablecloth in the first row is affected by ill-conditioned points. Furthermore, we

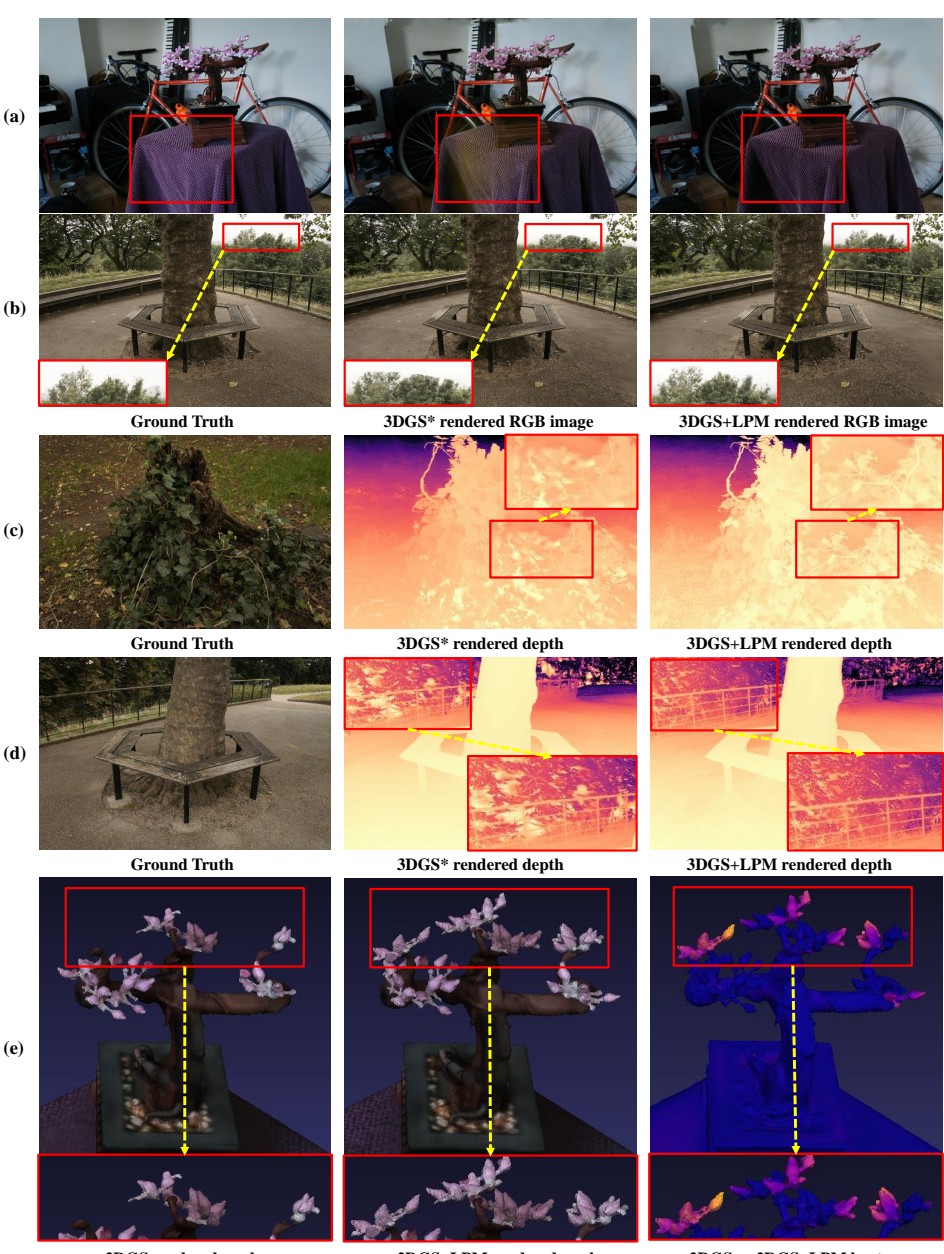

Figure 3: Qualitative evaluation of our LPM across diverse static datasets (Barron et al., 2022; Hedman et al., 2018). Our LPM improves 2DGS Huang et al. (2024) and 3DGS (Kerbl et al., 2023) on these challenging scenarios, e.g. (a) **Light effect**, (b) **Completeness in the distance**, (c, d) **Depth structure** and (e) **Mesh details**. See red patches for highlighted visual differences.

provide depth and mesh comparisons in the third and final rows. All these observations demonstrate that our geometry calibration with LPM offers an opportunity to correct these potentially ill-conditioned points, thereby enhancing the overall reconstruction accuracy.

**Results on dynamic 4D datasets** Table 2 presents a quantitative evaluation on the Neural 3D Video Dataset. Following established practices, training and evaluation are conducted at half resolution, with the first camera held out for evaluation (Li et al., 2022a). Integrating our **LPM** into SpaceTimeGS yields the best performance across all comparisons. Notably, our method demonstrates significant improvements in the challenging *Flame Salmon* scene compared to SpaceTimeGS

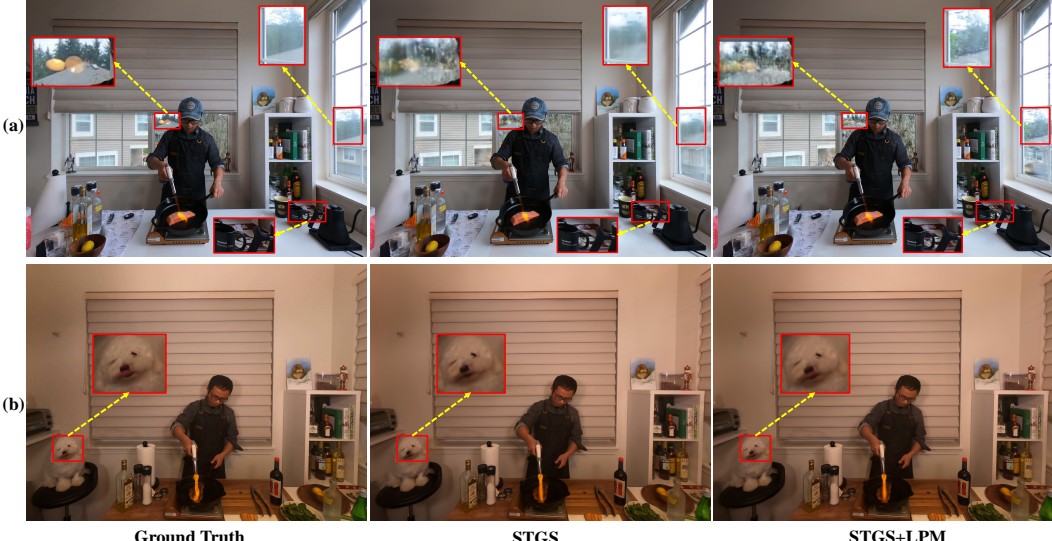

(a)

(b)

| Ground Truth | STGS | STGS+LPM |

Figure 4: Qualitative evaluation on dynamic Neural 3D Video dataset (Li et al., 2022b). LPM improves STGS (Li et al., 2023) for both scenes **Transparent** (*e.g.*, window) and **Dynamic movements** (*e.g.*, dog's tongue).

Table 2: Quantitative comparisons on the Neural 3D Video dataset. "FPS" is measured at a resolution of $1352 \times 1014$. Some methods only report results for a subset of scenes. For a fair comparison, we report LPM's results under two pre-existing settings. [1] Only includes the *Flame Salmon* scene. **Bold** represents best, underline indicates second best.

| Method | PSNR | DSSIM$_1$ | DSSIM$_2$ | LPIPS | FPS |
|---|---|---|---|---|---|
| LLFF [1] | 23.24 | - | 0.076 | 0.235 | - |
| DyNeRF [1] | 29.58 | **0.020** | 0.083 | 0.063 | 0.015 |
| Dynamic-4DGS [1] | - | - | - | - | 30 |
| 4DGS [1] | 29.38 | - | - | - | **114** |
| STGS [1] | 29.58 | 0.038 | 0.022 | 0.063 | 103 |
| STGS* [1] | 29.48 | 0.038 | 0.023 | 0.066 | 110 |
| STGS* [1] + **LPM** | **29.84** | 0.036 | **0.022** | **0.062** | 105 |
| StreamRF | 28.26 | - | - | **0.039** | 10.9 |
| NeRFPlayer | 30.69 | 0.034 | - | 0.111 | 0.05 |
| HyperReal | 31.10 | 0.036 | - | 0.096 | 2 |
| K-planes | 31.63 | 0.018 | - | 0.31 | 3 |
| MixVoxels-X | 31.73 | **0.015** | - | 0.064 | 4.6 |
| Dynamic-4DGS | 31.15 | - | 0.016 | 0.049 | 30 |
| 4DGS | 32.01 | - | - | 0.055 | 114 |
| STGS | 32.05 | 0.026 | 0.014 | 0.044 | 140 |
| STGS* | 31.99 | 0.026 | 0.015 | 0.045 | **145** |
| STGS*+ **LPM** | **32.40** | 0.025 | **0.014** | 0.045 | 140 |

(Li et al., 2023). Our approach not only surpasses previous methods in rendering quality but also maintains comparable rendering speed.

In addition to the quantitative assessment, we provide qualitative comparisons on the *Flame Salmon* and *Flame Steak* scenes, as illustrated in Figure 4. The quality of synthesis in both static and dynamic regions markedly outperforms STGS. Several intricate details, including the tree behind the window and the fine features like the dog's tongue, are faithfully reproduced with higher accuracy compared to STGS (Li et al., 2023). Both examples indicate that LPM improves upon STGS for superior scene modeling.

## 4.2 ABLATION STUDY

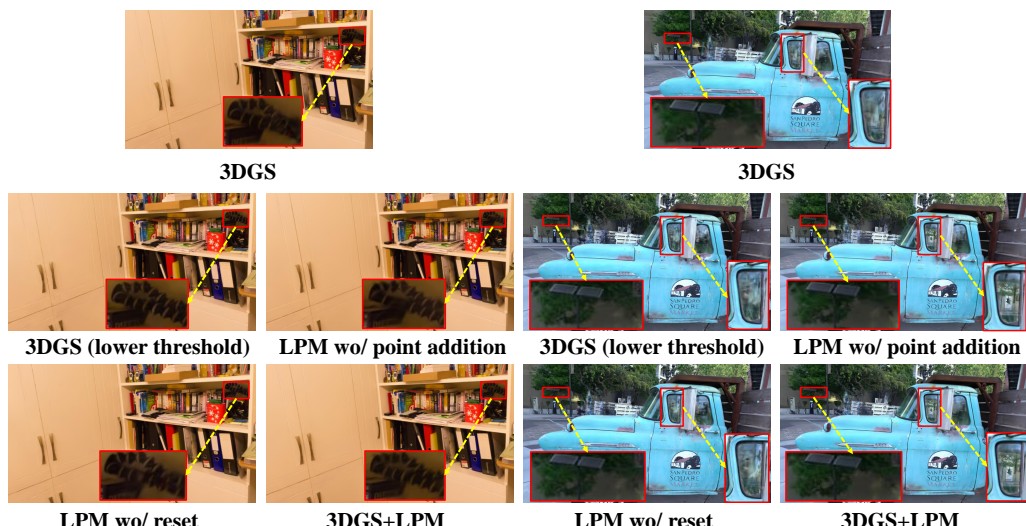

Figure 5: Effect of key operations of LPM. We show that the *point addition* operation effectively captures the geometric details in the scene; The *point reset* operation based on the error map further calibrate the geometry.

We conducted ablation studies on the more challenging scene: PlayRoom from Deep Blending (Hedman et al., 2018) and Truck from Tanks&Temples (Knapitsch et al., 2017).

Table 3: Cost-effectiveness analysis. Rendering speed of both methods are measured on our machine. **Note**: For 3DGS+LPM, training time includes the feature matching process.

| Scene | Method | PSNR | LPIPS | Gaussians | Training time |
|---|---|---|---|---|---|
| PlayRoom | 3DGS* | 30.03 | 0.244 | 232k | 22min |
| | 3DG* (lower threshold) | 29.69 | 0.240 | 523k | 36min |
| | GaussianPro | Out of Memory | | | |
| | PiexlGS | 30.09 | 0.241 | 186k | 35min |
| | 3DGS + **LPM** | 30.22 | 0.241 | 186k | 23min |
| Truck | 3DGS* | 25.42 | 0.146 | 257k | 19 min |
| | 3DGS* (lower threshold) | 25.45 | 0.127 | 635k | 35min |
| | GaussianPro | 25.40 | 0.164 | 312k | 36min |
| | PiexlGS | 25.51 | 0.121 | 518 | 37min |
| | 3DGS + **LPM** | 25.61 | 0.154 | 265k | 21min |

**Effectiveness and cost of LPM** We hypothesize that the Adaptive Density Control (ADC) tends to overlook under-optimized points due to its simplistic approach of thresholding the average gradient. The straight way to identify the all points is lowering threshold to densification process. Although this solution can reduce blurring in specific regions, such as the toy (red box) illustrated in Figure 5, it still has limitations. As shown in Table 3, lowering the threshold for 3DGS significantly increases the number of Gaussian points and decreases rendering speed. Additionally, the PSNR of the quantitative results decreases due to the introduction of unnecessary points in already dense areas. In contrast, *LPM* effectively generates points in areas indicated by the error map, leading to more accurate and detailed reconstructions while maintaining real-time rendering speed. As demonstrated by the qualitative comparison in Figure 5, 3DGS with LPM achieves superior qualitative results. We further compare our method with other recent methods that also focus on adaptive density control

(ADC). While PixelGS and GaussianPro achieve improvements in rendering quality, their training times increase substantially as they only consider point addition and extra gradient propagation. In contrast, LPM achieves a noticeable improvement with only a slight increase in training time due to (1) point matching (Lindenberger et al., 2023) is much faster (2) considering model expansion to dynamically prune the points by their addition number and (3) selecting points in error-contributing zones 3D zone using the parallel matrix operations.

**Individual points manipulation**    We study the effect of individual points manipulation of LPM, including the *point addition* and  *reset ill-conditional points*. The results in Table 4 show that, (1) each manipulation is useful with positive gain, suggesting that the  LPM is meaningful. (2) The *point addition* operation densify the under-optimized points which may be overlook in the 3DGS , further captures the geometry details (*e.g*., detail of toy and leaf of the tree, see Fig. 5). (3) Reset points in ceratin zone provide the opportunity of correct the ill-conditioned points to achieve geometry calibration, (*e.g*., window of the trunk, see Fig. 5).

Table 4: Performance comparison for different configurations

| Method | PlayRoom | | | Truck | | |
|---|---|---|---|---|---|---|
| | PSNR | LPIPS | SSIM | PSNR | LPIPS | SSIM |
| **Full LPM** | 30.22 | 0.241 | 0.910 | 25.61 | 0.154 | 0.883 |
| wo/ point addition | 30.10 | 0.241 | 0.910 | 25.43 | 0.153 | 0.883 |
| wo/ reset | 30.07 | 0.243 | 0.908 | 25.52 | 0.144 | 0.883 |

**Robustness to sparse training images**    We conducted further ablation studies to verify the impact of the number of training images. In Table 5, we present the results of training 3DGS and our method using randomly selected subsets comprising 25%, 50%, 75%, and 100% of the training images. Remarkably, our method consistently achieves superior rendering results compared to 3DGS across different percentages of training images.

Table 5: Effect of different training view ratios in the *PlayRoom* and *Truck*.

| Scene | Method | 25% | | 50% | | 75% | | 100% | |
|---|---|---|---|---|---|---|---|---|---|
| | | PSNR | LPIPS | PSNR | LPIPS | PSNR | LPIPS | PSNR | LPIPS |
| PlayRoom | 3DGS | 25.33 | 0.313 | 27.37 | 0.270 | 29.16 | 0.253 | 30.03 | 0.244 |
| | 3DGS+ **LPM** | 25.43 | 0.313 | 27.42 | 0.267 | 29.06 | 0.252 | 30.22 | 0.241 |
| Trunk | 3DGS | 22.46 | 0.177 | 24.15 | 0.154 | 24.86 | 0.150 | 25.42 | 0.146 |
| | 3DGS + **LPM** | 22.95 | 0.173 | 24.55 | 0.157 | 25.14 | 0.152 | 25.61 | 0.154 |

## 5    CONCLUSIONS AND LIMITATIONS

We propose Localized Point Management (LPM), a novel point management approach to address the limitations of the Adaptive Density Control (ADC) mechanism in 3D Gaussian Splatting (3DGS). The core idea of LPM is identifying the error-contributing 3D zones that require both point addition and geometry calibration under multiview geometry constraints, guided by image rendering errors. We implement appropriate operations for point densification and opacity reset. As a versatile plugin, LPM can be seamlessly integrated into existing 3DGS-based rendering methods. Extensive experiments across both static 3D and dynamic 4D scenes validate the efficacy of LPM in enhancing existing ADC mechanisms both quantitatively and qualitatively. While our method identifies the 3D Gaussian points that lead to rendering errors, it still follows the densification rules of 3DGS (Kerbl et al., 2023). This approach may not be optimal for under-optimized points, and we leave this aspect for further investigation.

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

# A APPENDIX

## A.1 ADDITIONAL RESULTS

**Per-scene Result of Static 3D**  We provide additional quantitative results for all three datasets in the tables referenced. Tables 6, 7, 8, 9, 10, and 11 present the metrics for each scene in the Mip-NeRF360 (Barron et al., 2021), Tanks&Temples (Knapitsch et al., 2017), and DeepBlending (Hedman et al., 2018) datasets. Our method consistently improve 3DGS (Kerbl et al., 2023) scene modeling in the vast majority of scenarios.

Table 6: Performance comparison of different methods on various scenes (PSNR ↑). (Part 1).

|  | Bicycle | Flowers | Garden | Stump | Treehill | Room |
|---|---|---|---|---|---|---|
| Plenoxels | 21.912 | 20.097 | 23.4947 | 20.661 | 22.487 | 27.594 |
| INGP-Big | 22.171 | 20.652 | 25.069 | 23.466 | 22.373 | 29.690 |
| Mip-NeRF 360 | 24.37 | 21.73 | 26.98 | 26.40 | **22.87** | **31.63** |
| 3DGS | 25.246 | 21.520 | 27.410 | 26.550 | 22.490 | 30.632 |
| 3DGS* | 25.166 | 21.576 | 27.388 | 26.637 | 22.487 | 31.53 |
| **3DGS + LPM** | **25.4** | **21.73** | **27.43** | **26.81** | 22.78 | 31.58 |

Table 7: Performance comparison of different methods on various scenes (PSNR ↑). (Part 2).

|  | Counter | Kitchen | Bonsai | Dr Johnson | Playroom | Truck | Train |
|---|---|---|---|---|---|---|---|
| Plenoxels | 23.624 | 23.420 | 24.669 | 23.142 | 22.980 | 23.221 | 18.927 |
| INGP-Big | 26.691 | 29.479 | 30.685 | 28.257 | 21.665 | 23.383 | 20.456 |
| Mip-NeRF 360 | **29.55** | **32.23** | **33.46** | 29.140 | 29.657 | 24.912 | 19.523 |
| 3DGS | 28.700 | 30.317 | 31.980 | 28.766 | 30.044 | 25.187 | 21.097 |
| 3DGS* | 28.90 | 31.43 | 32.14 | 29.08 | 30.03 | 25.42 | 21.91 |
| **3DGS + LPM** | 28.91 | 31.45 | 32.20 | **29.30** | **30.22** | **25.61** | **22.05** |

Table 8: Performance comparison of different methods on various scenes (LPIPS ↓). (Part 1).

|  | Bicycle | Flowers | Garden | Stump | Treehill | Room |
|---|---|---|---|---|---|---|
| Plenoxels | 0.506 | 0.521 | 0.3864 | 0.503 | 0.540 | 0.4186 |
| INGP-Big | 0.446 | 0.441 | 0.257 | 0.421 | 0.450 | 0.261 |
| Mip-NeRF 360 | 0.301 | 0.344 | 0.170 | 0.261 | 0.339 | 0.211 |
| 3DGS | 0.205 | 0.336 | **0.103** | **0.210** | **0.317** | 0.220 |
| 3DGS* | 0.211 | **0.336** | 0.107 | 0.215 | 0.324 | 0.218 |
| **3DGS + LPM** | **0.203** | 0.337 | 0.108 | 0.224 | 0.347 | **0.209** |

**Per-scene Result of Dynamic 4D**  In Table 13, we provide the PSNR on different scenes. The quanlitative results clearly show that LPM improve STGS (Li et al., 2023) to faithfully capture the subtle static and dynamic information.

# B MORE VISUALIZATIONS

Figure 6 provides more examples on static 3D and dynamic 4D dataset.

Table 9: Performance comparison of different methods on various scenes (LPIPS ↓). (Part 2).

|  | Counter | Kitchen | Bonsai | Dr Johnson | Playroom | Truck | Train |
|---|---|---|---|---|---|---|---|
| Plenoxels | 0.441 | 0.447 | 0.398 | 0.521 | 0.499 | 0.335 | 0.422 |
| INGP-Big | 0.306 | 0.195 | 0.205 | 0.352 | 0.428 | 0.249 | 0.360 |
| Mip-NeRF 360 | 0.204 | 0.127 | **0.176** | **0.237** | 0.252 | 0.159 | 0.354 |
| 3DGS | 0.204 | 0.129 | 0.205 | 0.244 | 0.241 | **0.148** | 0.218 |
| 3DGS* | 0.200 | 0.126 | 0.204 | 0.245 | 0.244 | 0.146 | 0.207 |
| **3DGS + LPM** | **0.200** | **0.125** | 0.202 | 0.241 | **0.241** | 0.154 | **0.209** |

Table 10: Performance comparison of different methods on various scenes (SSIM ↑). (Part 1).

|  | Bicycle | Flowers | Garden | Stump | Treehill | Room |
|---|---|---|---|---|---|---|
| Plenoxels | 0.496 | 0.431 | 0.6063 | 0.523 | 0.509 | 0.8417 |
| INGP-Big | 0.512 | 0.486 | 0.701 | 0.594 | 0.542 | 0.871 |
| Mip-NeRF 360 | 0.685 | 0.583 | 0.813 | 0.744 | 0.632 | 0.913 |
| 3DGS | 0.771 | 0.605 | 0.868 | 0.775 | **0.638** | 0.914 |
| 3DGS* | 0.765 | 0.606 | 0.867 | 0.773 | 0.634 | 0.920 |
| **3DGS + LPM** | **0.776** | **0.609** | **0.870** | **0.781** | 0.636 | **0.923** |

Table 11: Performance comparison of different methods on various scenes (SSIM ↑). (Part 2).

|  | Counter | Kitchen | Bonsai | Dr Johnson | Playroom | Truck | Train |
|---|---|---|---|---|---|---|---|
| Plenoxels | 0.759 | 0.648 | 0.814 | 0.787 | 0.802 | 0.774 | 0.663 |
| INGP-Big | 0.817 | 0.858 | 0.906 | 0.854 | 0.779 | 0.800 | 0.689 |
| Mip-NeRF 360 | 0.894 | 0.920 | 0.941 | 0.901 | 0.900 | 0.857 | 0.660 |
| 3DGS | 0.905 | 0.922 | 0.938 | 0.899 | 0.906 | 0.879 | 0.802 |
| 3DGS* | 0.908 | 0.927 | 0.942 | 0.901 | 0.907 | 0.882 | 0.815 |
| **3DGS + LPM** | **0.909** | **0.929** | **0.943** | **0.905** | **0.910** | **0.883** | **0.817** |

Table 12: Comparison of various methods across different scenes on the Mip-NeRF 360 dataset, Tanks&Temples and Deep Blending. 3DGS* indicates the retrained model from the official implementation. **Bold** represents best, underline indicates second best.

| Method | Indoor | | | Outdoor | | |
|---|---|---|---|---|---|---|
|  | PSNR | SSIM | LPIPS | PSNR | SSIM | LPIPS |
| 2DGS* | 24.210 | 0.705 | 0.282 | 30.105 | 0.911 | 0.211 |
| 2DGS + **LPM** | 24.427 | 0.716 | 0.264 | 30.432 | 0.919 | 0.193 |

Table 13: Performance comparison of different methods on various scenes (PSNR ↑).

|  | Coffee Martini | Spinach | Beef Cut | Salmon Flame | Steak Flame | Sear Steak |
|---|---|---|---|---|---|---|
| K-Planes-explicit | 28.74 | 32.19 | 31.93 | 28.71 | 31.80 | 31.89 |
| K-Planes-hybrid | 29.99 | 32.60 | 31.82 | 30.44 | 32.38 | 32.52 |
| MixVoxels | 29.36 | 31.61 | 31.30 | 29.92 | 31.21 | 31.43 |
| NeRFPlayer | **31.53** | 30.56 | 29.35 | **31.65** | 31.93 | 29.12 |
| HyperReel | 28.37 | 32.30 | 32.92 | 28.26 | 32.20 | 32.57 |
| Dynamic-4D | 27.34 | 32.46 | 32.90 | 29.20 | 32.51 | 32.49 |
| 4DGS | 28.33 | 32.93 | 33.85 | 29.38 | 34.03 | 33.51 |
| STGS | 28.61 | 33.18 | 33.52 | 29.48 | 33.64 | 33.89 |
| STGS* | 28.48 | 33.05 | 33.40 | 29.48 | 33.74 | 33.80 |
| **STGS+LPM** | 28.93 | **33.27** | **33.90** | 29.84 | **34.26** | **34.20** |

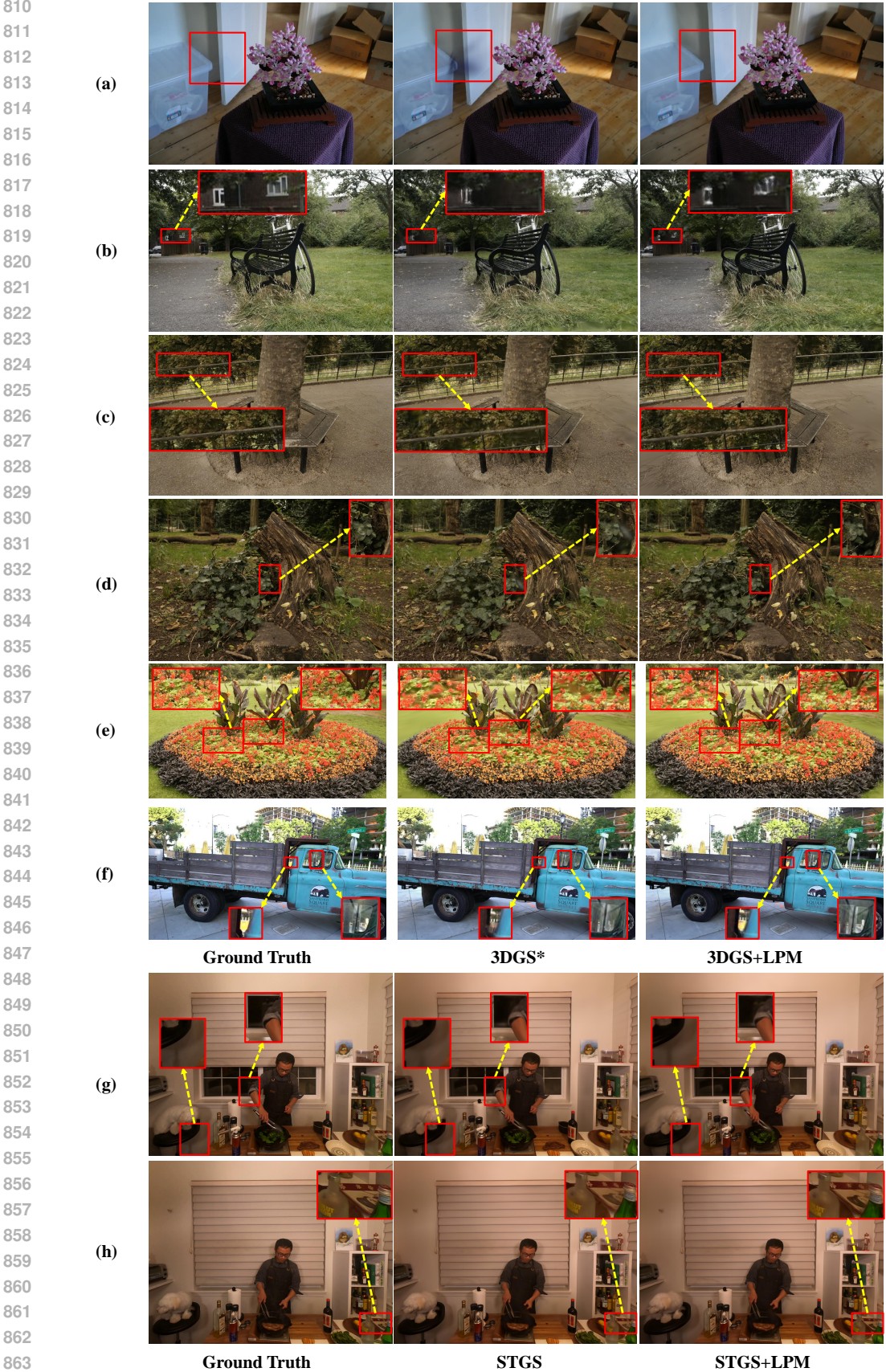

Figure 6: More qualitative comparisons on static 3D and dynamic 4D dataset.

