# OpenReview forum: "Improving Gaussian Splatting with Localized Points Management"
_ICLR.cc/2025/Conference — ICLR 2025 Conference Withdrawn Submission_

### Official Review · Reviewer_Fxkg · 2024-10-20

**Soundness:** 3
**Presentation:** 3
**Contribution:** 3
**Rating:** 6
**Confidence:** 4

**Summary:**

- This paper proposes a local point management (LPM) strategy to identify 3D regions that require point densification to alleviate the limitation of adaptive density control in 3D-GS.
- LPM can identify not only regions but also ill-conditioned points by leveraging multi-view geometric constraints and image rendering errors.
- LPM can be seamlessly integrated into existing 3D-GS-based methods. Experiments on static and dynamic scenes verify the proposed LPM's effectiveness.

**Strengths:**

1. The proposed LPM can identify 3D regions that cause incorrect rendering. For error regions, LPM densifies points or adds new Gaussians in these regions and resets the opacity of points in front of these regions.
2. By integrating LPM into existing 3D/4D GS methods, the rendering quality of static or dynamic scenes can be improved.

**Weaknesses:**

1. 3D zone identification requires the partial assignment predicted by LightGlue, which leads to some problems:
  - LPM cannot handle non-overlapping regions regions between two views.
  - Error and missing matches may harm LPM.

2. Although LPM is evaluated on Neural 3D Video dataset, for dynamic objects, the error region may move over time, and LPM lacks a mechanism to handle this situation. As shown in Figures 4 and 6, the improvement focuses on the static part.

3. Lacking some details. For example,
  - As shown in L202, how are the paired region adaptive adjustments?
  - The details formula of rCone in L208
  - What is the interval for applying LPM? Is that apply LPM every 100 iterations just like the densified interval in 3D-GS?

4. Lacking discussion about limitations and failure cases.

**Questions:**

See `Weaknesses`.

---

### Official Review · Reviewer_ytU9 · 2024-11-02

**Soundness:** 2
**Presentation:** 3
**Contribution:** 1
**Rating:** 5
**Confidence:** 4

**Summary:**

This paper introduces a point management strategy for 3DGS to identify error-contributing zones. Specifically, they use LightGlue to provide pair correspondence, thus solving regions that are incorrectly located. Experiment results show that their strategy can be applied to different 3DGS methods and slightly improve the performance.

**Strengths:**

* The proposed method is a plug-in module. Although it takes additional computation to use LightGlue, the performance seems to have improved.
* The motivation of the proposed method is intuitive.

**Weaknesses:**

* One crucial weakness is that the performance improvement by introducing such a module is minor. In most experiments, the PSNR is only improved by 0.1~0.2 PSNR, and the improvements on other metrics are even less noticeable, like SSIM. This raises the question of whether introducing such a module together with LightGlue is a good solution. In addition, as the performance difference could be due to randomness, an error-bound analysis would be helpful.
* Leveraging the pixel correspondence model may introduce additional errors since it may fail to find the correct correspondence. An analysis of such failure cases is helpful.

**Questions:**

* For 3D Gaussian splatting methods, the position of 3D Gaussians usually does not need to be perfect. Such flexibility allows the model to learn lightning and specular information. Thus, accurate 3D point management is less necessary, which may be the reason for the minor performance improvements in this paper. My question is whether this management helps more with the geometry modeling of 3DGS. Specifically, a quantitative comparison of surface modeling with 2DGS may help demonstrate its strength.

---

### Official Review · Reviewer_Crj3 · 2024-11-03

**Soundness:** 3
**Presentation:** 3
**Contribution:** 3
**Rating:** 5
**Confidence:** 4

**Summary:**

The paper introduces a point management method for 3D Gaussian Splatting, designed to improve point densification and correction. This approach leverages rendering errors in individual training views to identify 3D error zones by considering multi-view correspondences of 2D errors. Based on these detected error zones, point densification and opacity correction are applied to enhance the overall reconstruction quality. The results demonstrate promising improvements, with both quantitative and qualitative comparisons to the original 3D Gaussian Splatting and other prior methods.

**Strengths:**

The key contribution of the error-based point management technique sounds interesting and kind of novel, which seems naturally applicable to any GS-based representation and also leads to certain improvements over the standard technique used in 3DGS.

**Weaknesses:**

1. The method primarily relies on the assumption that regions with high errors require densification and correction. While this seems intuitively reasonable, it lacks a strong theoretical foundation, and many of the design choices appear ad-hoc without detailed mathematical explanations. Overall, the method shows some effectiveness, yet the mechanisms behind it remain unclear.

2. My main concern is on the quality. While the method offers some enhancement, the gains over standard 3D Gaussian Splatting appear incremental. Most quantitative results show PSNR improvements of less than 0.5 dB, with gains around 0.2 dB for static scenes, which is relatively marginal. Additionally, the visual comparisons reveal few notable differences, with only selected cropped examples—such as the truck windows—showing clearer improvement. These examples, however, are very few and appear carefully chosen. If the method specifically enhances quality in certain regions like transparent objects, this could be an interesting selling point, but a thorough explanation/evaluation, supported by more examples across various datasets, would be necessary to substantiate this claim.

**Questions:**

Overall, the method introduced in the paper has some novelty but I found it lacks a strong theoretical foundation. The quality improvement of 0.2~0.5db is also marginal. In general, while the approach offers an interesting advancement in point management for 3D Gaussian Splatting, it appears to be a relatively marginal step and still far from an optimal solution.

An additional comment: The paper "VET: Visual Error Tomography for Point Cloud Completion and High-Quality Neural Rendering" shares some similar insights with this work on leveraging visual errors for improvement in 3D point reconstruction and might be worth citing and discussing.

---

### Note · Authors · 2024-11-15

I have read and agree with the venue's withdrawal policy on behalf of myself and my co-authors.